# The Long-Term Effects of Budesonide Nasal Irrigation in Chronic Rhinosinusitis with Asthma

**DOI:** 10.3390/jcm11102690

**Published:** 2022-05-10

**Authors:** Seon Min Jung, Jin Hye Kwak, Moo Keon Kim, Kyung Tae, Seok Hyun Cho, Jin Hyeok Jeong

**Affiliations:** Department of Otolaryngology-Head & Neck Surgery, Hanyang University College of Medicine, Seoul 04763, Korea; smilingmini814@gmail.com (S.M.J.); kwak.jinhye@gmail.com (J.H.K.); for-kmg@hanmail.net (M.K.K.); kytae@hanyang.ac.kr (K.T.); shcho@hanyang.ac.kr (S.H.C.)

**Keywords:** nasal lavage, budesonide, sinusitis, asthma, nasal polyp

## Abstract

Chronic rhinosinusitis with nasal polyps (CRSwNP) in asthmatic patients has a high recurrence rate even after surgery. For this reason, oral steroids are frequently used, but their long-term use may cause side effects. The purpose of this study is to investigate the long-term effects of budesonide nasal irrigation (BNI) in CRSwNP and asthma. An analysis of 33 patients with CRSwNP and well-controlled asthma, who performed BNI for more than 12 months, was performed. We compared oral steroid and antibiotic dosages as well as nasal endoscopy scores before, and every six months after, BNI. The six-month dosages of oral steroids and antibiotics prescribed were significantly decreased at all time points after BNI compared to before BNI. When the dosages were compared at the time point immediately preceding six months, oral steroid intake decreased significantly until 12 months, and antibiotic intake decreased until 6 months. Furthermore, the endoscopic score decreased significantly until 12 months. The nasal symptom questionnaire score also significantly improved after BNI. Therefore, BNI is considered an effective treatment method that can improve subjective symptoms and objective intranasal findings while reducing oral steroid and antibiotic doses after long-term use in patients with CRSwNP accompanied by asthma.

## 1. Introduction

Chronic rhinosinusitis (CRS) is a chronic inflammatory disease, and the clinical course is influenced not only by a bacterial infection but also by various immune cells and their mediators [1,2]. CRS in asthmatic patients is a representative Th2-dominant CRS, and is often accompanied by nasal polyps [3,4].

Many cases of chronic rhinosinusitis with nasal polyps (CRSwNP) accompanied by asthma show a clinical course of recalcitrant rhinosinusitis, such as a low response to conservative treatment, a high recurrence rate even after surgery, and frequent exacerbation [5]. Therefore, steroids are frequently used to reduce repeated inflammation, but nasal topical steroid sprays are not effective when nasal polyps exist because it is difficult for the steroid to reach the target mucosa of the sinus [6]. Therefore, an oral systemic steroid is frequently used to treat repeated exacerbation and prevent recurrence [1,7]. However, long-term or frequent use of oral steroids carries a high risk of systemic side effects [8,9,10].

Recently, the efficacy of budesonide nasal irrigation (BNI) has been widely reported in patients with CRS [2,7]. Through this method, a large amount of steroids can effectively reach the entire nasal sinus and sinus mucosa at a higher pressure without systemic side effects.

In 2017, a prospective study was performed at our center, and we revealed the short-term effectiveness of BNI in patients with CRSwNP and asthma [11]. Therefore, the purposes of the present study were to investigate the long-term effects and usefulness of BNI in patients with CRSwNP and asthma and to find out the effective irrigation period for treatment.

## 2. Materials and Methods

### 2.1. Subjects

This study was conducted through a retrospective chart analysis of 33 patients with CRSwNP accompanied by well-controlled asthma who performed BNI for at least 12 months between 2014 and 2020. Among the 33 patients, 21 patients performed BNI after endoscopic sinus surgery (ESS), and 12 patients performed BNI without surgery. ESS was performed when the polyp was severe enough to completely block the middle meatus or spheno-ethmoidal recess (SER) so that at least the irrigation solution could enter the ostiomeatal unit and SER. ESS was not performed if there was enough space for the irrigation solution to enter the middle meatus and SER with small polyps. Therefore, the disease burden of patients who underwent ESS and those without ESS were similar, with a Lund-Kennedy (LK) endoscopy polyp score of less than 1.

To exclude the effect of the acute phase of surgery, BNI was started more than three weeks after the surgery, when the nasal mucosa was healed. Patients who did not undergo BNI continuously, or who needed to take oral steroids and immunosuppressants for other underlying diseases, were excluded. Patients attended regular follow-ups every three months in both otolaryngology for CRSwNP and the pulmonary medicine department for asthma. Patients with uncontrolled asthma who were not treated for asthma at our hospital pulmonary department, and one patient who needed oral systemic steroids for asthma control, were also excluded.

Oral steroids and antibiotics were prescribed according to the state of the patient’s disease at their outpatient visit. An oral steroid was prescribed when symptoms such as anosmia worsened or when the nasal polyp was severely exacerbated on the endoscopic examination. Oral antibiotics were prescribed for severe purulent rhinorrhea on the endoscopic examination.

This study was approved by the IRB (IRB No. 2021-06-014) and was conducted in compliance with the ethical standards of the responsible institution on human subjects as well as with the Helsinki Declaration.

### 2.2. Budenoside Nasal Irrigation

BNI was performed using a nasal washing bottle (250 mL; Nose sweeper bottle, Medicore Inc., Gyeonggi-do, Korea). An individually packaged 0.5 mg/2 mL budesonide suspension (Pulmican Respules^®^, Kuhnil Inc., Seoul, Korea) was mixed with 250 mL of normal saline. Half of the solution was used for each nasal cavity so that both nasal cavities could be irrigated evenly. After irrigation, oral gargle was performed with clean water to remove any solution remaining on the oral mucosa. BNI was performed once or twice a day depending upon the severity of nasal symptoms and status of endoscopic examination.

### 2.3. Clinical Parameters and Outcome Measures

To investigate the characteristics of CRS accompanied by asthma, paranasal sinus CT (PNS CT) scans and total immunoglobulin E (IgE) tests were performed before starting BNI. PNS CT scans were analyzed according to the Lund–Mackay CT score, which scores the degree of disease identified in each sinus.

To analyze the oral steroids dosage, the average of 6 months of total steroid dose (mg) taken before BNI, and the six-month total dosage of steroid taken every six months after BNI were recorded. Every six months, we assessed whether the steroid intake was decreased after BNI compared to the dose before BNI. In addition, the six-month dosage of oral steroids was compared to that of the immediately preceding six months to confirm the treatment time point that showed a progressive therapeutic effect, which was represented by a significant decrease in steroid dosage; this period was defined as the “cutoff point”.

The oral antibiotic intake was also compared using the same method of analyzing oral steroids. Since various types of oral antibiotics were prescribed, there was a limit to analyzing the antibiotic intake based on dose like steroids, so the antibiotic intake was analyzed by the number of days the antibiotic was taken.

To confirm the effect of BNI on the nasal mucosa and polyps, the nasal cavity was checked using endoscopy at outpatient visits every six months, and its state was scored according to the LK endoscopy score. The LK endoscopy score was also investigated and analyzed in the same way as mentioned above, before and every six months after BNI.

Finally, in order to identify the effect of BNI on patients’ subjective symptoms and quality of life, a 22-item Sinonasal Outcomes Test (SNOT-22) was performed, and the resulting score was compared between two time points: before irrigation and 12 months after irrigation.

### 2.4. Statistical Analysis

All statistical variables were expressed as mean ± standard deviation values. The period of statistical analysis was six months, and values of oral steroid usage, antibiotic usage, and endoscopic findings for each six-month period were analyzed using a paired *t*-test. Repeated-measures analysis of variance (ANOVA) was performed to confirm whether there was a continuous and significant decrease in all groups considering the characteristics of different follow-up periods. In addition, to identify the cutoff point for determining the effective therapeutic period of BNI, a paired *t*-test was performed. In all statistical analyses, a *p*-value of less than 0.05 was defined as statistically significant, using SAS version 9.4 (SAS Institute Inc., Cary, NC, USA).

## 3. Results

### 3.1. Clinical Parameters of Subjects

A total of 33 patients who underwent BNI for at least 12 months were enrolled, including 13 men and 20 women with an average age of 52.48 ± 14.41 years. BNI was performed for an average of 27.64 ± 7.07 months, using 18.48 ± 9.30 boxes (30 respules per box) of budesonide.

The patients’ total IgE values were 569.13 ± 620.21 IU/mL, and 27 patients showed a level higher than the normal level. Among the Lund–Mackay CT scores, the anterior ethmoid sinus had the highest score of 1.54 ± 0.65 points, which was significantly higher than that of other sinuses in the ANOVA test. A high total IgE level and a PNS CT finding of anterior ethmoid sinus predominance is consistent with the characteristics of CRSwNP with asthma.

### 3.2. Six-Month Dosage of Oral Steroid

The total amount of oral steroid intake (mg) was checked every six months (Figure 1). The six-month dosage of steroid showed a significant decrease statistically at all six-month time periods compared to before BNI (Table 1). In addition, repeated-measures ANOVA analysis was performed to confirm whether this trend of a decrease showed a significant difference in all groups with different consecutive periods of time, which also showed a significant difference (Table 2).

In order to confirm the cutoff point showing a significant therapeutic effect, we compared the total six-month dosage of steroids and that of the immediately preceding six months. A significant decrease in the steroid dose was confirmed up to 12 months after irrigation; after 12 months, there was a decreasing trend, albeit without statistical significance (Table 3).

### 3.3. Six-Month Dosage of Antibiotics

The total number of days of antibiotics taken over six months was investigated in the same manner as the oral steroids (Figure 2). The number of days of antibiotics taken was also significantly decreased at all six-month time points compared to before BNI (Table 4). When the cutoff point was analyzed, there was a significant decrease up to the first six months after irrigation, but, after that, there was no significant decrease compared to the previous six months.

### 3.4. LK Endoscopy Score

The LK endoscopy score was analyzed before, and every six months after, irrigation (Figure 3). The score was also improved at all six-month time points compared to before irrigation (Table 5). Analysis of the cutoff point showed a significant improvement in the LK score up to 12-month irrigation, but there was no significant improvement after that compared to the previous six months.

### 3.5. Nasal Symptoms Questionnaire (SNOT-22)

The patients’ subjective nasal symptoms were also improved. A significant improvement in the SNOT-22 score was obtained, from an average score of 40.03 ± 14.5 points before irrigation to an average score of 17.27 ± 10.68 points after irrigation (*p* < 0.001).

### 3.6. Clinical Side Effects

Only one patient felt mild intranasal irritation, but not to the extent that prevented BNI from being performed, and there were no other side effects related to steroid absorption during all periods.

## 4. Discussion

Asthma and CRS are closely related under the concept of “one airway, one disease.” In asthmatic patients, CRS is characterized by more nasal polyps, more severe symptoms, and worsening of nasal symptoms along with worsening of asthma [12,13]. Asthma has been mentioned as a representative risk factor of refractory rhinosinusitis [14,15]. In addition, patients with CRSwNP and asthma have a poor prognosis even after surgery. Zhang et al. reported that, even if ESS was performed extensively up to Draf 3 procedure, the five-year recurrence rate was high at 95.6% to 96.1% [15]. Gill et al. reported that the frequency of reoperation was twice as high as that in patients with asthma and, especially in those with nasal polyps, the frequency was six times higher [16]. For these reasons, steroids were frequently used. However, intranasal topical sprays are difficult to deliver effectively to the sinus mucosa in CRSwNP because of the polyps and polypoid mucosa [6]. 

The effect of topical steroids on the sinus mucosa is determined by the delivery technique, device, pressure, volume, position, and surgical status of the nasal cavity and sinus. A recent meta-analysis analyzed the effective distribution of drugs according to topical treatment methods in CRS patients and found that nasal irrigation is effective in delivering drugs to the nasal and sinus mucosae, especially in patients after surgery [17]. Accordingly, a method of BNI has been widely introduced in CRS and has already shown its usefulness [2,18,19]. In the study by Huang et al., when BNI was performed after ESS, not only the symptom scores for nasal obstruction, olfactory dysfunction, and rhinorrhea, but also the nasal endoscopic findings, were all improved compared to those in the saline irrigation group [2]. In addition, Harvey et al. compared the effects of steroid irrigation with a simple nasal spray in a double-blind, placebo-controlled randomized trial and reported that the subjective symptoms, endoscopic findings, and CT findings were significantly improved in the steroid irrigation group [19]. However, there has been no study that separately classified and analyzed the BNI effect in CRSwNP patients with asthma. 

Depending on their immunological characteristics of CRSwNP, patients’ clinical manifestations, prognosis, and treatment methods could be different; thus, the classification according to endotype reflecting each molecular immunodynamic characteristic has been widely used recently [1,20,21]. In the endotypic classification of CRSwNP, the Th2 immune response is usually activated and accompanied by asthma or aspirin sensitivity with eosinophilia [5,22]. CRS with a Th2 immune response is more refractory than that with a Th1 response; thus, local and systemic steroids are frequently used to control the Th2 immune response. From the endotypic classification point of view, it is not necessary to perform BNI in all CRSwNP cases; instead, it is preferable to perform BNI only if the Th2 immune response is related. This suggestion is supported by the study of Snidvongs et al. The SNOT-22 and the endoscopy scores showed significant improvements, especially in patients with a high tissue eosinophilia [23].

In 2017, our center revealed the short-term effect of six-month BNI in CRSwNP patients with asthma who frequently took oral steroids due to repeated exacerbations; both subjective symptoms and endoscopic findings were improved with a significant decrease in oral steroid intake for six months [11]. We tried to report the long-term effects of continuous BNI in this report; however, when BNI is performed for an extended period of six months or longer, there are concerns about side effects due to the absorption of a topical steroid and questions about how long it needs to be used to have a treatment effect.

There have been many studies on the safety of BNI, and most evaluated how the hypothalamic–pituitary–adrenal (HPA) axis or intraocular pressure could be affected by the topical steroid absorption. The safety of use of long-term BNI related to the HPA axis and intraocular pressure has already been confirmed in many studies [24,25]. However, in a paper by Soudry et al., some patients using budesonide for a long period showed a low level of stimulated cortisol in an ACTH-stimulation test, but there were no symptoms and it was normalized when BNI was stopped. Furthermore, when these patients continued irrigation under endocrine supervision, no further complications were observed [26]. 

Additionally, if long-term use is possible, another essential issue is how long BNI should be carried out. Our study was conducted to check for the “cutoff point” by comparing oral steroid doses, antibiotic doses, and endoscopic findings every six months with the previous six months to confirm this issue. Both oral steroid doses and endoscopic findings showed a cutoff point of 12 months, proving that performing BNI for about 12 months had a significant therapeutic effect compared to during the period before. Based on the results, performing BNI for about 12 months is recommended, during which time the nasal sinus mucosa is stabilized through remodeling by budesonide irrigation. It is thought that this should be elucidated in detail by further studies with histological and immunological examinations of the mucosa. Furthermore, if the mucosa stabilizes after irrigation for about one year and no symptoms manifest, then BNI could possibly be stopped. However, even if there is no significant improvement after irrigation for 12 months, the use of BNI can be extended while monitoring the symptoms of HPA-axis inhibition and intraocular pressure. 

In this study, most of the patients showed significant improvement after 12 months of continuous BNI. However, in 10 patients (30.3%), the course of the disease was so poor that they required more than 36 months of treatment. These patients can be regarded as cases of more severe recalcitrant CRS, which is not well responsive to BNI. For these patients, biologics, which have recently been developed and have expanding indications, may be available. Biologics have already begun to be used in refractory asthma and atopic dermatitis. Recently, they have been proven to be effective in CRSwNP, especially in the refractory cases associated with a Th2 immune response, like the subjects of our study [1,21,27]. However, there are limitations in the use of biologics because the indications are not clearly determined due to cost-efficacy aspects and potential risks. The European Position Paper on Rhinosinusitis and Chronic Rhinosinusitis (updated 2020) recommended using biologics in patients with CRSwNP [1,28]. Biologics was recommended to be used in severe cases of asthma-related Th2 inflammation with a lot of steroid intake. Based on the results of our study, it is thought that, along with the above recommendations, refractory CRSwNP patients with asthma who require continuous steroid use even after BNI for more than 12 months could consider the use of biologics. 

The significant decrease in antibiotic dosage up to six months after BNI suggests that the role of the nasal mucosa in a Th2 immune response is greater than the role of the bacterial infection in CRSwNP with asthma. When the nasal mucosa stabilizes and the nasal polyps decrease with 12 months of budesonide nasal irrigation, the ostiomeatal unit could remain open such that bacterial infection secretions do not easily occur, which results in a reduced role of antibiotics as a treatment option.

Although this study is a long-term report of a relatively large number of patients (33) with follow-up data collected to 36 months, it has several limitations. One of the limitations is that it is not a randomized controlled study. Another limitation is that the patient population was not uniform. If additional research is to be conducted, it is necessary to separate the operated and non-operated patients and to analyze them further according to the asthma control status. Finally, since BNI was stopped when the condition improved after 12 months, the number of enrolled patients decreased over the treatment period. As the number of subjects was reduced, it may appear that there was no statistical significance. 

In patients with CRSwNP accompanied by asthma with repeated exacerbations and recurrences, BNI can help reduce the need for oral systemic steroids and antibiotics by objectively improving the nasal condition and the patients’ subjective symptoms. When BNI was used for more than 12 months, many patients improved, and even after using it for up to 36 months, it was safely used without side effects. For those patients who do not improve after long-term budesonide nasal irrigation, the use of biologics can be considered.

## 5. Conclusions

In patients with CRSwNP accompanied by asthma with repeated exacerbations and recurrences, BNI can help reduce the need for oral systemic steroids and antibiotics by objectively improving the nasal condition and the patients’ subjective symptoms. 

When BNI was used for more than 12 months, many patients improved, and even after using it for up to 36 months, it was safely used without side effects. For those patients who do not improve after long-term budesonide nasal irrigation, the use of biologics can be considered.

## Figures and Tables

**Figure 1 jcm-11-02690-f001:**
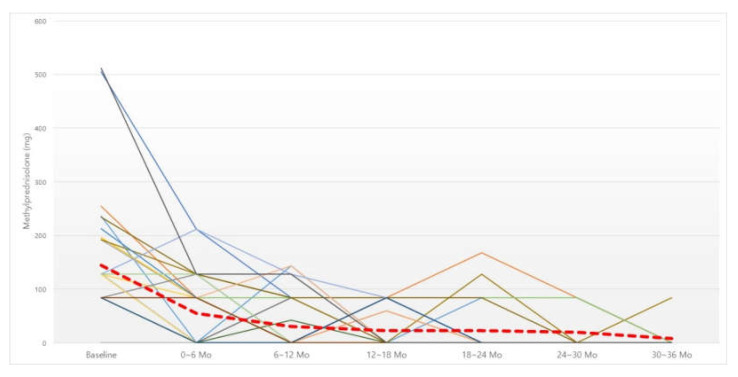
Changes in six-month dosage of oral steroid. Red dotted line, average of methylprednisolone intake (mg). Abbreviation: Mo, months.

**Figure 2 jcm-11-02690-f002:**
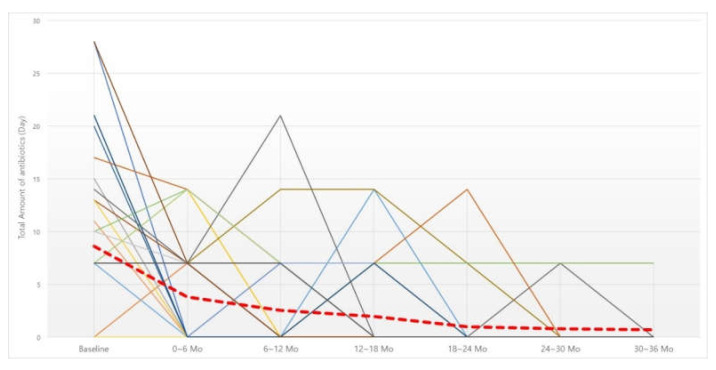
Changes in 6-month dosage of oral antibiotics. Red dotted line, average of oral antibiotics intake (day). Abbreviation: Mo, months.

**Figure 3 jcm-11-02690-f003:**
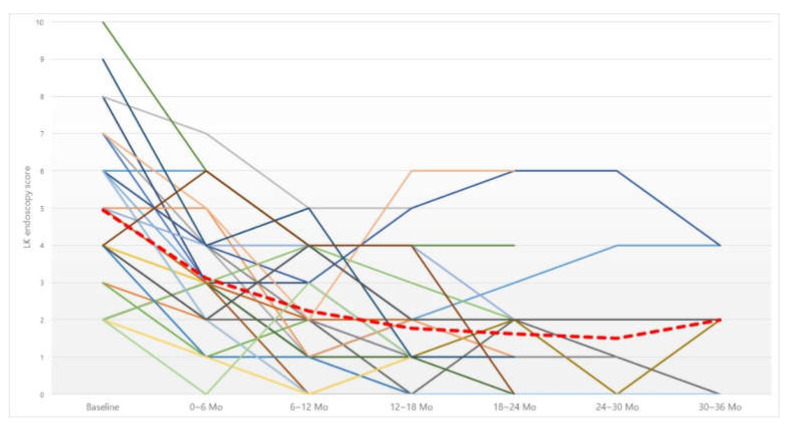
Changes in Lund–Kennedy (LK) endoscopy score. Red dotted line, average of LK endoscopy score. Abbreviation: Mo, months.

**Table 1 jcm-11-02690-t001:** Dosage of oral systemic steroid (methylprednisolone, mg) (paired *t*-test).

Baseline	0–6 Mo	6–12 Mo	12–18 Mo	18–24 Mo	24–30 Mo	30–36 Mo	*n*	*p*
145.05 ± 114.33	55.15 ± 64.94						33	<0.001
145.05 ± 114.33		30.48 ± 50.19					33	<0.001
146.95 ± 115.62			22.88 ± 37.37				32	<0.001
154.23 ± 120.22				22.57 ± 46.57			28	<0.001
185.36 ± 142.60					19.76 ± 36.73		17	<0.001
189.68 ± 137.76						8.40 ± 26.56	10	0.003

Abbreviation: Mo, months.

**Table 2 jcm-11-02690-t002:** Dosage of oral systemic steroid (methylprednisolone, mg) (repeated ANOVA test).

Baseline	0–6 Mo	6–12 Mo	12–18 Mo	18–24 Mo	24–30 Mo	30–36 Mo	*n*	*p*
145.05 ± 114.33	55.15 ± 64.94						33	<0.001
145.05 ± 114.33	55.15 ± 64.94	30.48 ± 50.19					33	<0.001
146.95 ± 115.62	54.25 ± 65.77	31.44 ± 50.69	22.88 ± 37.37				32	<0.001
154.23 ± 120.22	57.43 ± 66.54	35.93 ± 52.75	26.14 ± 38.93	22.57 ± 46.57			28	<0.001
185.36 ± 142.60	74.82 ± 74.36	50.71 ± 54.11	29.65 ± 41.38	37.18 ± 55.53	19.76 ± 36.73		17	<0.001
189.68 ± 137.76	55.20 ± 60.36	60.80 ± 56.15	33.60 ± 43.38	54.80 ± 63.02	25.20 ± 40.58	8.40 ± 26.56	10	<0.001

Abbreviation: Mo, months.

**Table 3 jcm-11-02690-t003:** Cutoff point of systemic oral steroid (methylprednisolone, mg) (paired *t*-test).

Baseline	0–6 Mo	6–12 Mo	12–18 Mo	18–24 Mo	24–30 Mo	30–36 Mo	*n*	*p*
145.05 ± 114.33	55.15 ± 64.94						33	<0.001 *
	55.15 ± 64.94	30.48 ± 50.19					33	0.029 *
		31.44 ± 50.69	22.88 ± 7.37				32	0.412
			26.14 ± 38.93	22.57 ± 46.57			28	0.696
				37.18 ± 55.53	19.76 ± 36.73		17	0.090
					25.20 ± 40.58	8.40 ± 26.56	10	0.343

* *p* < 0.05. Abbreviation: Mo, months.

**Table 4 jcm-11-02690-t004:** Dosage of oral antibiotics (days) (paired t-test).

Baseline	0–6 Mo	6–12 Mo	12–18 Mo	18–24 Mo	24–30 Mo	30–36 Mo	*n*	*p*
8.61 ± 8.32	3.82 ± 4.98						33	0.005
8.61 ± 8.32		2.55 ± 4.89					33	0.001
8.88 ± 8.31			1.97 ± 4.07				32	0.000
8.82 ± 8.46				1.00 ± 3.14			28	<0.001
8.18 ± 7.97					0.82 ± 2.32		17	0.001
6.90 ± 5.82						0.70 ± 2.21	10	0.008

Abbreviation: Mo, months.

**Table 5 jcm-11-02690-t005:** Lund–Kennedy endoscopy score (paired *t*-test).

Baseline	0–6 Mo	6–12 Mo	12–18 Mo	18–24 Mo	24–30 Mo	30–36 Mo	*n*	*p*
4.94 ± 2.21	3.12 ± 1.69						33	<0.001
4.94 ± 2.21		2.24 ± 1.58					33	<0.001
4.91 ± 2.23			1.78 ± 1.66				32	<0.001
5.19 ± 2.08				1.62 ± 1.68			26	<0.001
4.94 ± 1.63					1.50 ± 1.54		18	<0.001
4.80 ± 1.48						2.00 ± 1.33	10	<0.001

Abbreviation: Mo, months.

## Data Availability

Not applicable.

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
