# Peer review of "The Long-Term Effects of Budesonide Nasal Irrigation in Chronic Rhinosinusitis with Asthma"

_jcm, 2022, doi:10.3390/jcm11102690_

Round 1
Reviewer 1 Report
The authors used a heterogenic population which can affect the experimental outcome. The clinical course of patients with CRS with asthma is strongly related to the severity and acute exacerbations of asthmatic status. The patients need to be grouped according to the patient's status of asthma.
Author Response
Thanks for your precise comments. We fully agree with you. As we mentioned in the limitations, it is regrettable that the distribution of the patients was not uniform. However, a research setting was implemented to overcome the limitation.
First of all, all patients of our study did regular follow up every three months in both otolaryngology and pulmonary department. Under retrospective chart review, patients with uncontrolled asthma were excluded. Also, we excluded the patients who needed oral steroid for controlling asthma. Therefore, the patients in this study consist of a group of patients whose asthma is well controlled through regular follow-up and asthma treatment. We added the following to the material and method to clarify this as you point out: ‘Patients did regular follow-up every three months in both otolaryngology for CRSwNP and pulmonary medicine department for asthma. Patients with severe uncontrolled asthma and patients who needed oral systemic steroid for asthma control were also excluded.’
Secondly, this study is a long-term study of more than a year. Also, oral steroids and antibiotics doses were not recorded at a particular point in time, but considered the total doses during six-month. Therefore, we believe that these values ​​were relatively unaffected by an individual patient's asthma status at any point in time.
Of course, it is true that there are limitations that are not fully controlled. We look forward to doing further research with more homogeneous patients in the next study.
Reviewer 2 Report
The topic is very current.
The work is concise, concise.
The disease is very severe for patients and the possibility of improving the quality of life is significant as determined in the paper.
Literatures 57 % are in last 5 years, very good.
Author Response
Thanks for your kind comments. We also hope this study could be a good reference for the treatment of CRSwNP in asthmatic patients.
Reviewer 3 Report
The recurrence of nasal polyposis is extremely annoying for the attending physician and a reason for discouragement for the patient. Patients with nasal polyps and asthma are particularly prone to recurrence. For this reason, any effort to reduce the recurrence of polyps is commendable. This article is extremely interesting because it shows the long-term effects of nasal irrigation with budesonide in patients with CRSwNP and asthma. The study is extremely rigorous, the statistics are true, the methodology is clearly explained. extremely interesting discussions. The authors demonstrate that in patients with CRSwNP accompanied by asthma with repeated exacerbations and recurrences, budesonide nasal irrigations can help reduce the need for oral systemic steroids and antibiotics by objectively improving the nasal condition and the patients’ subjective symptoms.
Author Response
Thanks for you kind and precise comments. As you commented, CRSwNP is annoying to both patients and physicians, and affects patients' quality of life a lot. We also hope this study could be a good reference for the treatment of CRSwNP in asthmatic patients.
Round 2
Reviewer 1 Report
The authors replied as patients with uncontrolled asthma and who needed oral steroids for controlling asthma were excluded. It needs more explanation of how many patients were excluded by doing so. The authors claimed these exclusion criteria enhance the homogeneity of the patient population. If so the title should be changed to ~~ CRS with well-controlled asthma or ~~ CRS with asymptomatic asthma. I think the original title gives readers the wrong signal budesonide can be helpful to all asthmatic patients.
Author Response
We totally agree with your opinion that severity of asthma might influence the rhinosinusitis. However, we would appreciate it if you could generously understand that our study was conducted by comparing the scores for 6-month period, not one particular point, and excluding patients with poor asthma control or severe asthma. In addition, most of recently reported papers dealing with ‘Otolaryngologic treatment in CRSwNP with asthma’ didn’t categorize the patients according to the severity of asthma, so this study could not analyze the severity of asthma as well. In the further research, we will make sure to classify the patients more clearly according to the Asthma control test score.
The first point you pointed out was to report the number of people excluded by the exclusion criteria. Unfortunately, patients who do not control asthma in our hospital's pulmonary department cannot list them because they do not list them from the beginning, so the number cannot be known. Instead, we have described these points in more detail (line 64-67), and there was one case that oral steroids were used to control asthma, so this was specified (line 67).
Secondly, we would like to reply to the revision of the thesis title that you pointed out. In the paper, I think the title should concisely show what the topic of this paper is about, but the title does not have to elaborate on the inclusion criteria or the exclusion criteria or show the conclusion. If we add ‘well-controlled asthma’ or ‘asymptomatic asthma’ to the title, the reader may misunderstand that asthma is cured and has no problems at all. Our subjects are not patients who do not have any problems with asthma, but patients who visit the pulmonary department every three months and control asthma with step-up and step-down the medication depending on the degree of asthma. In addition, because the study does not include severe uncontrolled asthma, it is difficult to clarify, but from a pharmacological point of view, budesonide nasal irrigation may be effective in all asthma patients because it is thought to improve the nasal mucosa and polyp of CRS like intranasal steroid spray does. Therefore, instead of modifying the title, we add these details to the abstract (line 13), subjects (line 52), exclusion criteria (line 64-68), and limitation (line 301-303) in order to help readers understand accurately.
The effectiveness of budesonide nasal irrigation in CRS, CRS with NP, and CRS with asthma have already been shown in many papers. We would appreciate it if you could view this study as an additional study that shows the effectiveness of long-term use of budesonide and suggests the effective treatment period from the otolaryngology perspective.
We ask for your generous consideration for the shortcomings.
